# Evaluation of Cardiac Biomarkers and Expression Analysis of IL-1, IL-6, IL-10, IL-17, and IL-25 among COVID-19 Patients from Pakistan

**DOI:** 10.3390/v14102149

**Published:** 2022-09-29

**Authors:** Razi Ullah, Jadoon Khan, Nosheen Basharat, Danqun Huo, Ahmad Ud Din, Guixue Wang

**Affiliations:** 1Key Laboratory for Biorheological Science and Technology of Ministry of Education, State and Local Joint Engineering Lab for Vascular Implants College of Bioengineering Chongqing University, Chongqing 400030, China; 2Molecular Virology Laboratory, Department of Biosciences, COMSATS University, Islamabad 45320, Pakistan; 3Institutes for Systems Genetics, West China Hospital, Sichuan University, Chengdu 610212, China

**Keywords:** COVID-19, cardiac biomarkers, interleukins, Pakistan

## Abstract

Coronavirus disease 19 (COVID-19) is caused by viral infection of severe acute respiratory syndrome coronavirus 2 (SARS-CoV-2). Where upregulation of several important biomarkers and multiple organ dysfunction occurs, this study aimed to evaluate the association of cardiac biomarkers and CS induced acute lung damage with disease severity and mortality in survival of COVID-19 patients. A total of 500 COVID-19 patients with elevated cardiac biomarkers were studied for the analysis of myocardial abnormality through cardiac enzymes, inflammatory biomarkers, and the expression analysis of various cytokines, including *IL-1*, *IL-6*, *IL-10*, *IL-17*, and *IL-25* genes. The elevation of various cardiac enzymes including LDH (87%), CK (78.4%), TNI (80.4%), CK-MB (83%), and D-dimer (80.8%) were found correlated (*p* < 0.001) with COVID-19 infection. Cardiac enzyme elevation was highly associated with an increased level of inflammatory biomarkers such as CRP (14.2%), SAA (11.4%) and erythrocyte sedimentation rate (ESR) (7.8%) (*p* = 0.001 for all). The quantitative expression analysis of *IL-10*, *1L-17*, and *1L-25* were found to be high, while those of *IL-1* and *IL-6* were moderately elevated. The death-to-live ratio of COVID-19 patients was 457:43 indicating that the patients having elevated levels of both CKMB, D-dimer, CK and IL-1, IL-6, IL-10 and D-dimer, Troponin, CK and IL-1, IL-10 had high fatality rate (73% and 12% respectively). The current finding concludes that the evaluation of cardiac biomarkers with cytokine storm plays a significant role in COVID-19-associated anatomical organ damage, myocardial injury, and mortality. Physicians should pay special attention to cardiac biomarkers in patients with old age, inflammation, and comorbidities among COVID-19 infections.

## 1. Introduction

Coronavirus disease (COVID-19) is a global pandemic, infectious, and viral disease caused by the SARS-CoV-2 [1]. SARS-CoV-2 belongs to the genus Beta coronavirus, like MERS (Middle East Respiratory Syndrome) and SARS [2,3,4] A total of 12,236,677 confirmed cases of COVID-19 with 6,514,397 deaths have been reported worldwide [5]. COVID-19 is infectious with systemic infection that subsequently infects many organ systems; however, the lungs are the most significant and primary infecting organ. Several characteristics/factors, including preexisting comorbidities, immune status, and patient age, are involved in the severity of COVID-19 [3,6]. Generally, COVID-19 is responsible for acute respiratory distress syndrome and lung injury; however, cardiac injury associated with this multisystemic viral infection is yet the topic of debate.

The adverse effects of various systemic viral infections such as SARS [7] and influenza [8] on the cardiovascular system have been reported since the 1930s [9]. The development of severe symptoms among COVID-19 patients among individuals with preexisting coronary artery disease has been previously observed in China from the literature of the Centers for Disease Control and Prevention (CDC) [5]. The disease severity among intensive care unit (ICU) admitted with previous cardiovascular complications was three times more severe than non-ICU patients [10]. However, evidence of myocardial injury caused by COVID-19 has not yet been reported [11,12]. The acute COVID-19 cardiovascular syndrome (AcovCS), i.e., myocardial injury caused by COVID-19, might be due to increased myocardial demand in response to the stress of infection, endothelial dysfunction, and platelets activation due to the thrombogenic environment produced by increased cytokines production as well as direct myocardial damage [2,3]. The over-expression of several cardiac biomarkers associated with cardiac injury in several reports [13,14,15] show a linkage, while several others show a lack of association [16,17].

The first step of the defense mechanism is the activation of the innate immune arm during viral infections. The successful recognition of genomic RNA or DNA of viruses through PRRs (pattern recognition receptors) found on dendritic cells of the host leads to the production of various chemokines and cytokines [18], which attracts other immune cells such as T-cells, neutrophils, and macrophages to recruit to the infection site based on their source and targeted cells [19]. The cytokine storm (CS), which is the sudden elevation of various cytokines causing active inflammation, leads to critical conditions like acute respiratory distress syndrome (ARDS) and multiorgan failure-associated death [16]. The important pro-inflammatory cytokines such as TNF-α (Tumour Necrosis Factor-α) and interleukins (IL), including IL-6 and IL-1, are critical players in the initial immune response against infection. However, anti-inflammatory cytokines, including IL-10, are responsible for maintaining homeostasis in the immune response during sustained infection [20]. The increased synthesis of both anti-inflammatory and pro-inflammatory cytokines in COVID-19 patients has been reported previously [21,22,23,24]. A recent comprehensive meta-analysis reported the enhanced level of Interferon-γ (IFN-γ), TNF-α, IL-10, IL-8, IL-6, IL-4, IL-2R and IL-2, whereas no significant increase in IL-17 and IL-1β level among severe COVID-19 patients [25].

Keeping the above literature in view, it is of utmost importance to evaluate the association between CS-associated acute lung damage and cardiac biomarkers and their impact on the survival and death of COVID-19 patients. So, the current study evaluated the cardiac biomarkers and expression analysis of IL-1, IL-6, IL-10, IL-17 and IL-25 among COVID-19 patients from Pakistan.

## 2. Materials and Methods

### 2.1. Study Design and Ethical Consideration

The current retrospective cohort study evaluated cardiac biomarkers and various cytokines among COVID-19 patients. Molecular diagnosis of COVID-19 patients followed by its classification as the severe or common type was made according to interim guidelines of the World Health Organization. Ethical approval was granted by the Ethical Review Board of COMSATS University, Islamabad, Pakistan.

### 2.2. Inclusive Criteria and Control Group

Patients initially positive for SARS-CoV-2 through RT-PCR were included in this study. The studied control individuals had similar demographic and physiological characteristics (age, gender, residence, etc). However, they were non-infected for various common viral (HBV, HCV, HIV, Corona) and metabolic disorders such as (diabetes, CVD, CPD, and Hypertension).

### 2.3. Exclusive Criteria

Patients who were negative for COVID-19 RT-PCR, and missing medical information were excluded from the current study

### 2.4. Blood Samples and Data Collection

A total of 500/793 patients were included in the study. At the same time, others with missing medical information, CK, LDH, CK-MB, TNI and D-dimer laboratory results, and duplicated records were excluded. Approximately 10cc of blood from each patient in DNA/RNA Shield Blood Collection Tube (cat#R1150, Zymo Research, 2911 Dow Ave., Tustin, CA 92780, USA) was collected. Clinical and demographic data were collected from the hospital patients’ records, including name, age, sex, clinical symptoms, and underlying comorbidities. Other laboratory tests such as complete blood count (CBC), Liver functional tests (LFTs), renal functional tests (RFTs), random blood sugar, lipids profile, inflammatory and cardiac biomarkers (CK-MB (creatine kinase-myocardial band), LDH (lactate dehydrogenase), CK (creatine kinase), D-dimer and TNI (troponin I) were recorded and compared. Serum TNI level above the 99th-percentile upper reference limit (>26.2 μg/L) was used for cardiac injury evaluation. The clinical observation with death and discharge were critically monitored, and patients were categorized according to the absence and presence of myocardial injury and cardiac biomarkers abnormality.

### 2.5. Extraction and Preparation of RNA from PBMCs

The isolation of Peripheral Blood Mononuclear Cells (PBMCs) was carried out on Ficoll–Paque solution (Amersham Biosciences AB Björkgatan 30, SE-751 84 Uppsala, Sweden) through density gradient centrifugation following the standard procedure of the manufacturer. According to the manufacturer’s instructions, total RNA was extracted from PBMCs by RNX™-Plus reagent (SinaClon, Tehran, Iran). The removal of genomic DNA was done by treating the extracted RNA with DNase I (Fermentas, Lithuania) at 37°C for 15 min, followed by verification through spectrophotometry and 1% agarose gel electrophoresis.

### 2.6. Synthesis of cDNA and Genes Expression Analysis

Copied DNA (cDNA) was synthesized through reverse transcription of RNA with Random Hexamer primers (MWG, Fraunhoferstr. 22 D-82152 Martinsried, Germany) and Oligo dT through RevertAid^™^ M-MuLV RT (Fermentas, Lithuania) as per manufacturer’s guidelines. Quantitative Real-Time Polymerase Chain Reaction (qRT-PCR) for selected gene expression through Step One Applied Biosystems PCR system (Applied Biosystem/MDS SCIEX, Foster City, CA, USA) was performed by using Primer set for each of FoxP3 [26]. Complement of differentiation (*CD4*, *CD25*) [27], *IL10* were analyzed, and Glyceraldehyde 3-phosphate dehydrogenase (*GAPDH*) [28] was used as an internal control. The final reaction volume of 20 μL containing 4 μL of 5X EvaGreen^®^ qPCR Mix Plus (ROX) (Solis BioDyne, Teaduspargi 9, 50411 Tartu, Estonia), 10 ng of cDNA template, and 4 pM of each forward and reverse primers. The entire samples were run in triplicate, and the normalized expression was used for data analysis. Determination of normalized expression and calculation of ΔCt values were done by subtracting the average Ct value of *GAPDH* from the average Ct value of *IL-10*, *CD4, CD25*, and *FoxP3*. Relative expression was calculated by 2^−ΔΔCt^ formula as described [29]. The thermal reaction conditions are shown in Table 1.

### 2.7. Statistical Analysis

The entire data was reviewed and entered in SPSS (Ver. 16) (IBM Corp., Armonk, NY, USA) and GraphPad (version 7.00, San Diego, CA, USA) for statistical analysis. Percentage (%) was used for categorical variables, while mean (standard deviation (SD) and median (IQR interquartile range) were used for normal and nonnormal distribution of continuous variables. The difference and association between myocardial injury and enzymes with COVID-19 were analyzed using Mann–Whitney U or χ^2^ test and One Way ANOVA. Pearson’s correlation analysis and univariable logistic regression analysis were used for binary variables to explore the risk factors associated with myocardial enzyme abnormality and myocardial injury. Multivariable and univariable logistic regression models explored the risk factors associated with in-hospital death.

## 3. Results

### 3.1. Clinical and Demographic Characteristics

Cardiac biomarkers including CK-MB, LDH, CK or TNI were hyperregulated among 481/500 (96.2%) compared to 19/500 (3.8%) patients within the normal range. The ratio of severe to non-severe COVID-19 cases were 467/33 (93.4%:6.6%). The significantly observed common symptoms include cough (303, 60.6%) followed by fever (297, 59.4%) and chest pain/tightness (218, 43.6%). Older age was significantly associated with abnormal cardiac enzymes (>49 years; *p* < 0.001); however, male patients dominated females (70:30; *p* = 0.009). Furthermore, the abnormality in cardiac biomarkers 18.2% (*p* = 0.005) was more commonly observed in COVID-19 patients showing two or more comorbidities. The common comorbidities observed among COVID-19 in the current study was chronic pulmonary disease (5.2%; *p* = 0.044), diabetes (11.4%; *p* = 0.002) and hypertension (22.6%; *p* = 0.01). The entire clinical and demographic observations are presented in Table 2.

### 3.2. Laboratory Parameters

Various blood parameters, including white blood count (15.2%), neutrophil count (18.4%), and lower lymphocyte count (20.4) (all *p* < 0.001), were significantly associated with abnormal cardiac biomarkers containing COVID-19 patients. C-reactive protein (CRP) among 14.2% individuals, Erythrocyte Sedimentation Rate (ESR) (7.8%), and SAA (11.4%) (all *p* < 0.001) were elevated inflammatory biomarkers among hyperregulated cardiac enzymes linked COVID-19 patients. Elevated cardiac biomarkers including CK-MB (83; *p* = 0.016), TNI (80.4%; *p* < 0.001), CK (78.4%; *p* < 0.001), D-dimer (80.8%; *p* < 0.001) and LDH (87%; *p* < 0.001) were found strongly associated with higher levels of urea nitrogen (*p* = 0.011), γ-transgluta-minase (*p* < 0.001), aspartate aminotransferase (AST) (*p* < 0.001) and alanine aminotransferase (ALT) (median 36 vs. 21.5 U/L; *p* = 0.001) as shown in Table 2.

### 3.3. Correlation of Cytokines with Cardiac Biomarkers

Analysis of various clinical parameters with respect to cardiac biomarkers shows a strong correlation (*p* < 0.05), as shown in Figure 1.

### 3.4. Risk Factors for COVID-19 Associated Death in Hospitalized Patients

The death to live/discharge ratio among the abnormal cardiac biomarker associated COVID-19 patients were 91.4:8.6% or 457:43. Most of the deaths were observed in patients with older age (*p* < 0.001), rate of respiration (*p* =0.001), myocardial injury patients (*p* < 0.001), patients with comorbidities (*p* = 0.003), levels of urea nitrogen (*p* = 0.001), direct bilirubin (*p* = 0.049) and total bilirubin (*p* = 0.030), AST (*p* = 0.009), ALP (*p* = 0.004), SAA (*p* = 0.034), ESR (*p* = 0.017), procalcitonin (*p* = 0.017), CRP (*p* = 0.001), LDH (*p* = 0.001), CK (*p* = 0.003), TNI levels (*p* = 0.014), platelets (*p* = 0.001), neutrophils (*p* = 0.001), white blood cells (*p* = 0.001) and IL-6 (*p* = 0.004) as shown in (Table 3).

### 3.5. Expression Analysis of IL-1, IL-6, IL-10, IL-17 and IL-25 Genes

The quantitative expression analysis of *IL-1, IL-6, IL-10, IL-17*, and *IL-25* genes revealed that *IL-1* and *IL-6* had moderately high while *IL-10, IL-17* and *IL-25* were highly expressed, which were significantly correlated with COVID-19 infection (*p* < 0.05) (Figure 2).

## 4. Discussion

Severe Acute Respiratory Syndrome-Coronavirus-2 (SARS-CoV-2) can attack and infect many vital organs of the human host [14]. SARS-CoV-2-associated cardiac injury is mainly responsible for an increased mortality rate among COVID-19 patients [34]. The availability of limited literature on the degree of myocardial injury caused by COVID-19 leads to uncertainty in their prognosis and clinical outcomes. That is why the current retrospective study was designed to evaluate the association of cardiac biomarkers and cytokine storm (CS) induced acute lung damage with disease severity and mortality or survival of COVID-19 patients.

A recent study reported that elevated levels of myohemoglobin, CK-MB, hs-TNI and older age was strongly correlated with the severity of COVID-19 infection [35] Another study found cardiac injury among 7.2% of COVID-19 patients [13], such as myocardial injury among 27.8% of those COVID-19 patients having elevated cardia troponin T level [36] A previous study reported an increased level of LDH among 16.0% and CK-MB among 10.8% of severe acute respiratory syndrome (SARS) patients in 2003 in Guangzhou institute of respiratory diseases, China [37]. This study observed that 467/500 (93.4%) of COVID-19 patients have abnormal cardiac enzymes, and 457 exhibited myocardial injury, parallel to the study of Zhang et al. [37] from China. The rate of mortality, disease severity, Novel Coronavirus Pneumonia (NCP), Alkaline Phosphatase (ALP), Aspartate Aminotransferase (AST), TNI, Serum Amyloid A (SAA), Erythrocyte Sedimentation Rate (ESR) and C-reactive protein (CRP) were highly elevated among the patients with abnormal cardiac enzymes, corresponding with other reports where one or several factors were significantly enhanced [13,14,15,37]. A comprehensive report declared cardiovascular diseases a potential risk factor for COVID-19 infection [14,17] The literature, as mentioned earlier, suggested that abnormal cardiac biomarkers and myocardial injury may lead to severe outcomes among COVID-19 patients. The elevation in CK, LDH and cardiac TNI levels among COVID-19 patients works as a death warrant, which needs proper investigation and special attention of clinicians [37]. The above-mentioned literature suggested that COVID-19 associated myocardial injury might be caused due to initiation of a severe inflammatory response. Therefore, it is necessary to give exceptional and early attention to anti-inflammatory therapy during clinical observation and hospital admission [38].

The severity of COVID-19 is also correlated with several risk factors such as former heart diseases, kidney diseases and old age [14]. The current study observed high infectivity and severity of COVID-19 among the old age population, as reported by [14,37]. In the case of SARS and MERS-associated infections and death, old age was a significantly associated factor [17,39,40]. It has been suggested that most older patients are immune tolerant, and the T-lymphocytes are not properly working inflammatory cytokines such as TNF-α and IL-6 are hyperregulated and lead to cytokine storm [41,42]. Being responsible for the development of severe pneumonia. All this phenomenon potentially contributes to unsatisfactory outcomes among old-age patients infected with COVID-19 compared to young individuals [37].

The exact mechanism of COVID-19-associated cardiac injury is yet the topic of research; however, the association between inflammation and severe clinical outcomes among COVID-19 patients, viral load versus clinical symptoms progression might support the hypothesis of COVID-19-derived exaggerated inflammatory response [16,43]. The expression of both IL-1β and IL-6 might be responsible for this inflammatory process [16,43,44,45,46]. The current study observed hyperregulation of IL-1 and IL-6 among COVID-19 infected individuals as observed previously [47]. Elevated activity of IL-1β and IL-6 cytokines has been observed at the nasopharynx region, which is the key site of infection where higher bioactivity of IL-6 is strongly correlated with a higher load of SARS-CoV-2. The finding mentioned above suggests that IL-6 mediated inflammation is associated with the presence of SARS-CoV-2 antigens. However, we could not conclude that IL-6 mediated inflammation may continue in the tissues during the latter stages of severe COVID-19, when the viral load ends [43]. An enhanced cytokine response has been detected among COVID-19 isolated leukocytes and whole blood during transcriptome analysis as well as from the tissues of nasopharyngeal origin compared to the control population [48]. The literature, as mentioned earlier, suggests that IL-6 and IL-1β play a prominent role in the pathogenesis and severity of COVID-19, and that is why the IL-6 and IL1β antagonists are in a clinical trial [49]. Henceforth, the cytokine-modulating therapeutic approaches are now in clinical trials of COVID-19 patients. The initiation of inflammatory response activates the various immune cells, further enhancing the inflammation and gene expression during COVID-19 responsible for severe outcomes in patients [50].

The hyperregulation of anti-inflammatory cytokines IL-10 has been observed in patients of severe COVID-19 infection [16,45]. Being multifunctional nature, the primary function of IL-10 is to control the inflammatory response. In contrast, it has been known for introducing non-responsiveness or energy of T-cells in anti-tumor cell response [51] and during viral infection [52]. The removal of IL-10 on a genetic level or blockade of its receptors or through anti-IL-10 antibodies leads to the elimination of viral infection [52,53,54] or bacterial pathogen [55]. The current observation of the hyper-regulation of IL-10 gene among COVID-19 patients supported the finding that elevated levels of IL-10 in severe COVID-19 patients were initially ascribed to the negative feedback through its anti-inflammatory activities [56]. Furthermore, the hyper-regulation of IL-10 was significantly linked with an increased risk for future cardiovascular and coronary atherosclerosis [57].

An elevated level of IL-17 has been observed in the serum of human influenza-infected patients [58,59] and COVID-19 [60]. Accordingly, an elevated level of serum IL-17 among COVID-19 was observed in this study also. In contrast to our study, the downregulation of tissue IL-17A/IL-8 and lower neutrophil scores among COVID-19 patients have been reported earlier [4]. The neutrophil reduction observed in COVID-19 patients might support the immune evasion of the virus and continuous survival in the patients [61,62]. However, IL-17 might enhance the destruction of lung parenchyma through inadequate recruitment of neutrophils, hyper-regulation of pro-inflammatory mediators responsible for the prevention of apoptosis through expression of granulocyte-colony stimulating factor during acute respiratory distress syndrome (ARDS) [24,63].

A hyper-regulation of IL-17E, also known as IL-25, among COVID-19 patients was observed in the current study, produced by all epithelial cells [64,65,66]. As per previous reports, IL-17E can actively participate in T-helper type 2 immune responses in the lungs and gut [64,66,67,68] however, recent reports observed that IL-17E/IL-25 actively participates in several inflammatory disorders of the skin [69,70]. Besides its new role in COVID-19 patients, the IL-25 possesses unique structures and can regulate immune and inflammatory responses [71] and angiogenesis [72]. Further evidence showed that IL-25 promotes cell proliferation, inhibits apoptosis [73], and affects the development of fibrosis and hypoxic-ischemic injury [6,74], tumor metastasis [75] and metabolism processes [76,77]. IL-17 and IL-17E (IL25) were also highly elevated.

## 5. Conclusions

It is concluded from the current finding that the elevation of cardiac biomarkers and inflammatory markers with upregulation of several cytokines have a significant role in COVID-19 associated anatomical organ damage, myocardial injury, and mortality. Cardiac Biomarkers should be appropriately investigated soon after COVID-19 hospitalization, and proper attention should be given to inflammatory markers with cardiac enzymes, especially those with older age and other comorbidities. There is a need of further confirmation of the various pro-inflammatory and anti-inflammatory cytokines; however, due to lack of resources, we were not able to proceed further. The non-availability of resources for western blot confirmation was one of the study’s main limitations. The evaluation of cytokines expression inside a suitable animal model will be a valuable addition.

## Figures and Tables

**Figure 1 viruses-14-02149-f001:**
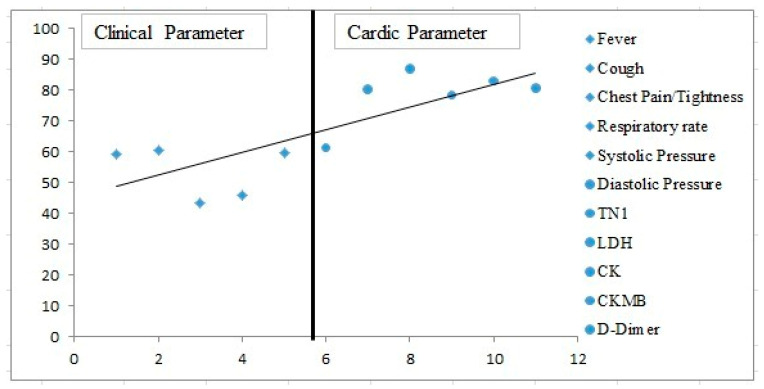
Scatter plot showing correlation among clinical parameters and cardiac biomarkers.

**Figure 2 viruses-14-02149-f002:**
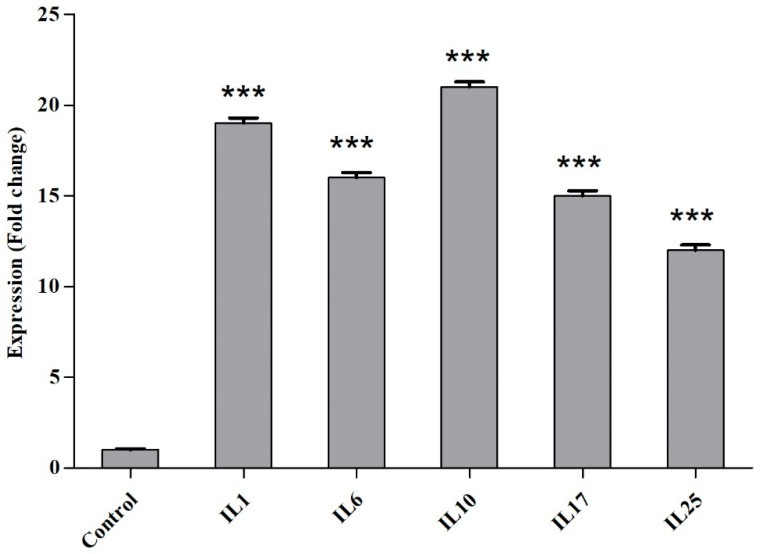
Immune genes expression in COVID-19 patients and control group (*** significantly different than control (*p* < 0.05).

**Table 1 viruses-14-02149-t001:** RT-PCR Primers and Thermocycler conditions.

Primer	Primer Sequence	Annealing	Reference
*IL-1*	F-5′ TCT GGGGCATACTCACAGGGGT-3′ R-5′ AGCTGGGTTGTGGTAGCCTTACTG-3′	52 °C/30 s	[30]
*IL-6*	F-5′ CATGTTCTCTGGGAAATCGTGG-3′ R-5′ AACGCACTAGGTTTGCCGAGTA-3′	51.4 °C/30 s	[28]
*IL-10*	F-5′ACTACTAAGGCTTCTTTGGGAA-3′ R-5′ CAGTGCCAACTGAGAATTTGG-3′	56 °C/30 s	[31]
*IL-17*	F-5′ GCTCCTCTGGCCTTGATGATG-3′ R-5′ CTTGTACGTCCACGGCGAGATG-3′	53 °C/30 s	[32]
*IL-25*	F-5′ GTGCCTGTGCCTCCCCTA-3′ R-5′ CGC CTGTAGAAGACA GTCTG-3′	57 °C/30 s	[33]
*GAPDH*	F-5′ CTTTGTCAAGCTCATTTCCTGG-3′ R-5′ TCTTCCTCTTGTGCTCTTGC-3′	57 °C/30 s	[28]

**Table 2 viruses-14-02149-t002:** Clinical, Demographic and Laboratory Parameter of COVID-19 admitted Patients.

Clinical and Demographics Parameters	Normal Value	Total Samples (500)	Abnormal Cardiac Biomarkers (*n*= 481)	Normal Cardiac Biomarkers (*n* = 19)	Z/χ^2^	*p*-Value
**Age Groups**						
Age group 1 (18–28)		95	65 (13%)	30 (6%)	5.870	<0.001
Age group 2 (29–38)		89	73 (14.6%)	16 (3.2%)		
Age group 3 (39–48)		96	89 (17.8%)	7 (1.4%)		
Age group 4 (49–58)		116	91 (18.2%)	25 (5%)		
Age group 5 (58 and above)		104	88 (17.6%)	16 (3.2%)		
**Gender and Symptoms**						
Male		351	236 (47.2%)	115 (23%)	6.676	0.008
Female		149	107 (21.4%)	42 (8.4%)		
Fever		413	297 (59.4%)	116 (23.2%)	0.009	0.926
Cough		382	303 (60.6%)	79 (15.8%)	0.261	0.61
Chest pain/tightness		295	218 (43.6%)	77 (15.4%)	0.303	0.582
Respiratory rate		273	229 (45.8%)	44 (8.8%)		0.004
Systolic pressure	90–140 mm Hg	327	299 (59.8%)	28 (5.6%)	−3.046	0.002
Diastolic pressure	60–90 mm Hg	358	307 (61.4%)	51 (10.2%)	−0.027	0.978
**Comorbidities**						
Hypertension		181	113 (22.6%)	68 (13.6%)	6.655	0.01
Diabetes		95	57 (11.4%)	38 (7.6%)	9.571	0.002
Other CVD		83	63 (12.6%)	20 (4%)	0.398	0.528
CPD		56	26 (5.2%)	30 (6%)	4.058	0.044
With more than one comorbidity		139	91 (18.2%)	48 (9.6%)	7.865	0.005
**Laboratory Observations**						
Blood routine tests		high				
White blood cells	3.5–9.5 10^9/^L	201	76 (15.2%)	125 (25%)	3.59	<0.001
Neutrophil	1.8–6.3 10^9/^L	232	92 (18.4%)	140 (28%)	5.282	<0.001
Lymphocyte	1.1–3.2 10^9/^L	185	102 (20.4%)	83 (16.6%)	−5.25	<0.001
Platelet count	125–350 10^9/^L	255	97 (19.4%)	157 (31.4%)	−0.188	0.851
Haemoglobin	130–175 g/L	49	31 (6.2%)	18 (3.6%)	−1.62	0.105
Monocyte count	0.1–0.6 10^9/^L	352	165 (33%)	187 (37.4%)	−0.799	0.424
**Random Blood Sugar and Lipids**						
FBG	4.1–5.9 mmol/L	189	92 (18.4%)	97 (19.4%)	3.185	0.001
Total cholesterol	<5.2 mmol/L	316	200 (40%)	116 (23.2%)		0.22
Triglyceride	<1.7 mmol/L	281	196 (39.2%)	85 (17%)	−0.591	0.555
HDL	1.16–1.42 mmol/L	132	69 (13.8%)	63 (12.6%)	−2.091	0.036
LDL	2.7–3.1 mmol/L	199	115 (23%)	84 (16.8%)		0.18
**Cytokines and Inflammatory Biomarkers**						
CRP	<8 mg/L	152	71 (14.2%)	81 (16.2%)	−6.588	<0.001
Procalcitonin	<0.5 μg/L	183	101 (20.2%)	82 (16.4%)	−5.242	<0.001
SAA	<10 mg/L	93	57 (11.4%)	36 (7.2%)	5.326	<0.001
ESR	<20 mm/h	117	39 (7.8%)	78 (15.6%)	5.598	<0.001
IL-1	0–5 pg/mL	431	157 (31.4%)	274 (54.8%)	−4.201	<0.001
IL-6	0.12–2.9 ng/L	465	186 (37.2%)	279 (55.8%)	−4.912	<0.001
IL-10	0.1–5 ng/L	472	135 (27%)	337 (67.4%)	−4.832	<0.001
IL-17	≤10 pg/mL	416	162 (32.4%)	254 (50.8%)	−4.710	<0.001
IL-25	23 and 200 pg/mL	422	179 (35.8%)	243 (48.6%)	−4.347	<0.001
**Cardiac Biomarkers**						
CK	138–174 U/L	438	392 (78.4%)	46 (9.2%)	−5.743	<0.001
LDH	109–245 U/L	481	435 (87%)	46 (9.2%)	−9.962	<0.001
TN1	<26.2 μg/L	450	402 (80.4%)	48 (9.6%)	−5.207	<0.001
CKMB	0–24 μg/L	476	415 (83%)	61 (12.2%)	−2.404	0.016
D-dimer		487	404 (80.8%)	83 (16.6%)		
**Renal and Liver Functional Tests**						
ALT	5–40 U/L	236	160 (32%)	76 (15.2%)	−3.466	0.001
AST	8–40 U/L	179	101 (20.2%)	78 (15.6%)	−5.522	<0.001
ALP	40–150 U/L	153	89 (17.8%)	64 (12.8%)	−2.122	0.034
γ−Transglutaminase	11–50 U/L	211	176 (35.2%)	35 (7%)	−4.053	<0.001
Total bilirubin	5.1–19 μmol/L	189	132 (26.4%)	57 (11.4%)	−3.307	0.001
Urea nitrogen	2.9–8.2 mmol/L	103	67 (13.4%)	36 (7.2%)	−2.557	0.011
Creatinine	46–92 μmol/L	197	130 (26%)	67 (13.4%)	−1.754	0.079
Uric acid	149–369 μmol/L	156	102 (20.4%)	54 (10.8%)	−1.055	0.292
**Clinical Typing and Outcome**						
Severe		467	461 (92.2%)	6 (1.2%)	41.314	<0.001
Non−severe		33	22 (4.4%)	11 (2.2%)		
Death		457	441 (88.2%)	16 (3.2%)		<0.001
Discharge		43	16 (3.2%)	23 (4.6%)		

**Table 3 viruses-14-02149-t003:** Risk Factors for COVID-19 Associated Death.

Clinical Parameters	*p*-Value	OR/Univariate Analysis (95% CI)
Old age	0.002	1.201 (1.03–1.132)
Gender	0.341	0.618 (0.23–1.664)
Rate of Respiration	0.003	1.231 (1.213–1.341)
Tightness/Chest pain	0.712	1.31 (0.551–3.483)
Non-severe vs. Severe	0.005	2.621 (1.342–5.117)
Underlying comorbidities	0.001	5.121 (1.809–13.531)
Platelets	0.001	0.977 (0.966–0.988)
Neutrophil	0.001	1.415 (1.207–1.658)
WBC	0.001	1.315 (1.142–1.514)
IL-6	0.004	1.018 (1.006–1.031)
IL-10	0.060	1.229 (0.991–1.524)
TN1	0.014	1.009 (1.002–1.017)
LDH	0.001	1.004 (1.002–1.007)
CK	0.003	1.002 (1.001–1.004)
EGFR	0.001	0.952 (0.932–0.972)
SAA	0.034	1.003 (1–1.005)
ESR	0.017	1.031 (1.006–1.057)
CRP	0.001	1.031 (1.018–1.045)
AST	0.009	1.007 (1.002–1.013)
ALT	0.683	0.998 (0.987–1.009)
ALP	0.004	1.014 (1.004–1.023)
Procalcitonin	0.017	1.654 (1.093–2.504)
Creatinine	0.085	1.004 (0.999–1.009)
UREA nitrogen	0.001	1.714 (1.339–2.149)
Total bile acid	0.158	1.066 (0.976–1.165)
Direct bilirubin	0.049	1.069 (1–1.142)
Total bilirubin	0.030	1.062 (1.006–1.121)
γ-Transglutaminase	0.500	1.002 (0.995–1.01)

## Data Availability

All the collected relevant research data is presented and included in this paper; however, further inquiries can be directed to the corresponding authors.

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
