# Peer review of "Evaluation of Cardiac Biomarkers and Expression Analysis of IL-1, IL-6, IL-10, IL-17, and IL-25 among COVID-19 Patients from Pakistan"

_viruses, 2022, doi:10.3390/v14102149_

Round 1

Reviewer 1 Report

Dear Authors,

The manuscript entitled “Evaluation of Cardiac Biomarkers and Expression Analysis of 2 IL-1, IL-6, IL-10, IL-17, and IL-25 among COVID-19 Patients from Pakistan” has been thoroughly reviewed. Though Ullah et al designed, performed, and written very well, it looks as preliminary and incomplete study. The study analyses cardiac biomarkers and cytokines expression in patients with SARS-COV2 infection. However, there are some major and minor concerns those should be clarified:

·         Authors should mention details for controls. Did the authors perform statistical analysis for the calculation of the sample size? Please mention it in the materials and methods?

·         Were these COVID-19 patients on medication or a fresh COVID-19 diagnosed patients were included in the study?

·         Did the authors follow up the patients to analyze the recovery parameters and compare with pre- and post-COVID-19 symptoms, and biomarkers and cytokine expression?

·         The study lacks the molecular mechanism that should investigate on how and why expression of cytokines was found in COVID-19 patients. In my opinion, the study would be of great interest to the readers if this mechanism could be revealed.

·         Pro- and anti-inflammatory cytokines were analyzed in the current study; however, it lacks to correlate the expression of cytokines among pro- and anti-inflammatory molecules and with biomarkers and other detected parameters. That should be analyzed. Moreover, it would be interesting to know the correlation. Add this analysis also.

·         Authors have studied the cytokine expression only at RNA levels; however protein levels of these cytokines should be analyzed that would make it with strong evidences.

·         Check the tables to mention/correct names of genes/cytokines.

·         Figure 1 should show with statistical analyses.

·         Authors should include limitation of the study.

·         It is advised to correct English errors and typographic errors throughout the manuscript.

·         All the abbreviations must be defined at the first place (e.g. CS) where it is written first.

Author Response

We appreciate the time taken by the editorial board and reviewers involved in the improving of this manuscript entitled “Evaluation of Cardiac Biomarkers and Expression Analysis of IL-1, IL-6, IL-10, IL-17, and IL-25 among COVID-19 Patients from Pakistan”. We have tried our best to make changes and correction based on reviewers and editorial board recommendations. Reviewers’ comments have been colored black and response to reviewers have been colored red. In the main text file, the modified portions have been Colored in colored.   

Reviewer Comments

Reviewer 1. 

  • Authors should mention details for controls. Did the authors perform statistical analysis for the calculation of the sample size? Please mention it in the materials and methods?

Answer:  Thank you very much for critical evaluation of the manuscript. The studied control individuals were those having similar demographic and physiological characteristic (age, gender, residence, etc) however they were non-infected for various common viral (HBV, HCV, HIV, Corona) and metabolic disorders like (diabetes, CVD, CPD, Hypertension). Manuscript (Line 108 to 111).

  • Were these COVID-19 patients on medication or a fresh COVID-19 diagnosed patients were included in the study?

Answer: The entire study included fresh COVID-19 positive patients.

  • Did the authors follow up the patients to analyze the recovery parameters and compare with pre- and post-COVID-19 symptoms, and biomarkers and cytokine expression?

Answer: The entire symptoms and the biomarkers were recovered/become normal in 98.9% of the survived patients except those 1.1% having elevated level of CKMB that’s why they were not mentioned in the study.

  • The study lacks the molecular mechanism that should investigate on how and why expression of cytokines was found in COVID-19 patients. In my opinion, the study would be of great interest to the readers if this mechanism could be revealed.

Answer: Yes, I agree with the reviewer recommendations, however due to lack of facilities and financial problems we can’t proceed in cell lines and proper animal model, and these are included in the limitation section of this manuscript. Manuscript (Line 329-333)

  • Pro- and anti-inflammatory cytokines were analyzed in the current study; however, it lacks to correlate the expression of cytokines among pro- and anti-inflammatory molecules and with biomarkers and other detected parameters. That should be analyzed. Moreover, it would be interesting to know the correlation. Add this analysis also.

Answer: As per the Reviewer # 2 suggestions; the correlation analyses have been performed for all cytokines with cardiac biomarkers and were evaluated through and represented in the scatter plot (Fig #1). Manuscript (Line 193-198)

  • Authors have studied the cytokine expression only at RNA levels; however, protein levels of these cytokines should be analyzed that would make it with strong evidence.

Answer: Thank for you this valuable advice. Suggested part has been adjusted accordingly. If we see in the Table # 1, the expression analysis of various cytokines through ELISA represent the expression on protein level; they are also represented in the bar graph again for visibility purposes (Fig#.1) Confirmation through western blot was the main limitation of the study mentioned in the Manuscript and have been added in limitation section (Line 329-333).

  • Check the tables to mention/correct names of genes/cytokines.

Answer: The entire table was rechecked for the correction of genes/cytokines names.

  • Figure 1 should show with statistical analyses.

Answer: The Figure 1 (Now it become Figure 2) was represented with statistical analysis

  • Authors should include limitation of the study.

Answer: the limitation of the study has been included in the Manuscript (Line 329-333).

  • It is advised to correct English errors and typographic errors throughout the manuscript.

Answer: The paper was reviewed properly, and the typographic and English errors were corrected accordingly

  • All the abbreviations must be defined at the first place (e.g. CS) where it is written first.

Answer: The entire abbreviations were defined at the first place throughout the manuscript

Reviewer 2 Report

Evaluation of Cardiac Biomarkers and Expression Analysis of IL-1, IL-6, IL-10, IL-17, and IL-25 among COVID-19 Patients from Pakistan

Razi Ullah, Jadoon Khan, Nosheen Basharat, Danqun Huo, Ahmad Ud Din *, Guixue Wang *

Here, Ullah et. al is reporting the association between the cardiovascular syndrome (CS) associated acute lung damage and cardiac biomarkers and expression analysis of IL-1, IL-6, IL-10, IL17 and IL-125 among COVID 19 patients from Pakistan.

Here are my concerns regarding the manuscript

1.    The only experimental proof the authors show in the manuscript is the expression validation of IL-1, IL-6, IL-10, IL-17 and IL-25. In the experimental setup the patients from COVID-19 were compared to the control.  However, in the methods section the authors did not mention anything about these samples. Which samples were that? Where are these patients from? Is there any additional information about the patients? 

2.    I would strongly suggest that the authors should use more methods to validate the elevation of the markers such as western blot of 5 samples from patients and control patients. I would also recommend ELISA of patients to compare with the rt-qPCR.

3.    Authors should include the correlation plots with respect to LDH, CK, D-dimer and TNI and to the clinical parameters mentioned in table 3.

4.    Authors should also describe the expression data in terms of fold change.

5.    Authors needs to conclude and discuss the results precisely in the discussion section.

If the authors provide all the information listed above, then it would make a good story.

Author Response

We appreciate the time taken by the editorial board and reviewers involved in the improving of this manuscript entitled “Evaluation of Cardiac Biomarkers and Expression Analysis of IL-1, IL-6, IL-10, IL-17, and IL-25 among COVID-19 Patients from Pakistan”. We have tried our best to make changes and correction based on reviewers and editorial board recommendations. Reviewers’ comments have been colored black and response to reviewers have been colored red. In the main text file, the modified portions have been Colored in colored.   

Reviewer 2

Evaluation of Cardiac Biomarkers and Expression Analysis of IL-1, IL-6, IL-10, IL-17, and IL-25 among COVID-19 Patients from Pakistan

Razi Ullah, Jadoon Khan, Nosheen Basharat, Danqun Huo, Ahmad Ud Din *, Guixue Wang *

Here, Ullah et. al is reporting the association between the cardiovascular syndrome (CS) associated acute lung damage and cardiac biomarkers and expression analysis of IL-1, IL-6, IL-10, IL17 and IL-125 among COVID 19 patients from Pakistan.

Here are my concerns regarding the manuscript

  1. The only experimental proof the authors show in the manuscript is the expression validation of IL-1, IL-6, IL-10, IL-17 and IL-25. In the experimental setup the patients from COVID-19 were compared to the control. However, in the methods section the authors did not mention anything about these samples. Which samples were that? Where are these patients from? Is there any additional information about the patients? 

Answer:  Thank you very much for critical evaluation of the manuscript. The studied control individuals were those having similar demographic and physiological characteristic (age, gender, residence, etc) however they were non-infected for various common viral (HBV, HCV, HIV, Corona) and metabolic disorders like (diabetes, CVD, CPD, Hypertension). Manuscript (Line 108 to 111).

  1. I would strongly suggest that the authors should use more methods to validate the elevation of the markers such as western blot of 5 samples from patients and control patients. I would also recommend ELISA of patients to compare with the rt-qPCR.

Answer: Yes, we agree with the recommendation of reviewer about the validation of samples through Western blot however due to lack of facilities and financial problems we were not able to do so. Mentioned in the limitation of the manuscript (Line 329-333). Also, if we see in the Table # 1, the expression analysis of various cytokines through ELISA representing the expression on protein level; and in the bar graph again for visibility purposes (Fig#.1).

  1. Authors should include the correlation plots with respect to LDH, CK, D-dimer and TNI and to the clinical parameters mentioned in table 3.

Answer: As per both Reviewer # 1 and 2 the correlation of all cytokines with cardiac biomarkers were evaluated through and represented in the scatter plot (Fig #.1). Manuscript (Line 193-198).

  1. Authors should also describe the expression data in terms of fold change.

Answer: The data is represented in terms of fold changes (Figure # 2) in the manuscript (Line 213-217).

  1. Authors needs to conclude and discuss the results precisely in the discussion section.

Answer: The discussion section has been modified accordingly.

Round 2

Reviewer 1 Report

Dear Authors,

Revised manuscript has substantially been improved and answered all queries. However, the manuscript is still required some minor editing such as gene names were written as IL-6, IL17 through out the MS.

Author Response

Author's Reply to the Review Report (Reviewer 1)

Concern. The revised manuscript has substantially been improved and answered all queries. However, the manuscript is still required some minor editing, such as gene names being written as IL-6, IL17 throughout the MS.

Response: Changes have been made and errors found have been corrected as suggested. Changes can be found in MS track changes file.

Reviewer 2 Report

Hello, 

The paper have improved with additional two experiments showing the scatter plot and the expression level of genes. This adds in to the information but I would still suggest to work on the presentation of the paper. 

Figure 2 is compressed and cannot be read properly.

Author Response

Author's Reply to the Review Report (Reviewer 2)

Concern: The paper have improved with additional two experiments showing the scatter plot and the expression level of genes. This adds in to the information but I would still suggest to work on the presentation of the paper.

Figure 2 is compressed and cannot be read properly.

Response. We have improved the MS as suggested, and that specific figure has been changed to be able to read properly.